# The Impact of Oral Health Behaviors, Health Belief Model, and Absolute Risk Aversion on the Willingness of Japanese University Students to Undergo Regular Dental Check-Ups: A Cross-Sectional Study

**DOI:** 10.3390/ijerph192113920

**Published:** 2022-10-26

**Authors:** Ichiro Sumita, Naoki Toyama, Daisuke Ekuni, Takayuki Maruyama, Aya Yokoi, Daiki Fukuhara, Yoko Uchida-Fukuhara, Momoko Nakahara, Manabu Morita

**Affiliations:** 1Department of Preventive Dentistry, Faculty of Medicine, Dentistry and Pharmaceutical Sciences, Okayama University, 2-5-1 Shikata-Cho, Kita-Ku, Okayama 700-8558, Japan; pj0p7nno@s.okayama-u.ac.jp (I.S.); dekuni7@md.okayama-u.ac.jp (D.E.); t-maru@md.okayama-u.ac.jp (T.M.); yokoi-a1@cc.okayama-u.ac.jp (A.Y.); de20041@s.okayama-u.ac.jp (D.F.); pric37ll@s.okayama-u.ac.jp (M.N.); mmorita@md.okayama-u.ac.jp (M.M.); 2Department of Oral Morphology, Faculty of Medicine, Dentistry and Pharmaceutical Sciences, Okayama University, 2-5-1 Shikata-Cho, Kita-Ku, Okayama 700-8558, Japan; de20006@s.okayama-u.ac.jp

**Keywords:** oral health behaviors, health belief model, absolute risk aversion

## Abstract

Oral health behaviors, risk aversion, and the health belief model are associated with health behaviors. However, there have been few studies that investigated the association between these factors and the willingness to undergo regular dental check-ups. The purpose of this cross-sectional study was to investigate the associations between the willingness of Japanese university students to undergo regular dental check-ups and oral health behaviors, the health belief model, and absolute risk aversion. An analysis was conducted with the cooperation of questionnaire respondents (*n* = 748) who underwent dental check-ups at Okayama University. The students answered questionnaires on oral health behaviors, the health belief model, absolute risk aversion, and willingness to undergo regular dental check-ups. The logistic regression analysis showed significant positive associations (*p* < 0.05) between oral health behaviors (use of the inter-dental brush and the dental floss) and the health belief model with the willingness to undergo regular dental check-ups. However, there was no significant association with absolute risk aversion (*p* > 0.05). These results suggest that willingness to undergo regular dental check-ups was associated with oral health behaviors and the health belief model, but not with absolute risk aversion.

## 1. Introduction

Oral health behaviors, including regular dental check-ups, are important to maintain oral health and prevent oral disease. Undergoing regular dental check-ups is significantly associated with good oral hygiene status in Japanese university students [1,2]. However, the percentage of Japanese students undergoing regular dental check-ups is 12.3–30.2% [1,2,3], lower than the general Japanese population according to the National Health and Nutrition Survey in 2016 (52.9%) [4]. On the other hand, in Japan, an oral examination once a year is mandatory for students until the high school period. Thereafter, regular dental check-ups are the individual’s responsibility. Thus, it is very important to encourage university students to undergo regular dental check-ups.

A health belief model proposed by Rosenstock (1966) is a well-known theoretical model that explains health behavior and focuses on the individual’s beliefs related to health behavior [5]. The health belief model is spelled out in terms of the following 4 constructs: perceived susceptibility, severity, benefits, and barriers. These concepts were proposed as accounting for people’s “readiness to act [6]”. Other constructs were added later, including “cues to action” (events, people, or things that are associated with change in behavior) and “self-efficacy” (one’s confidence to act) [7]. Factors in the health belief model influence the promotion or reduction of oral health behaviors [8,9]. However, few past studies have shown a relationship between the willingness to undergo regular dental check-ups and the health belief model.

The concept of risk aversion has been used in diverse fields including economics, behavioral economics, and psychology [10] and risk aversion refers to the tendency of an economic agent to strictly prefer certainty to uncertainty. In other words, those with high-risk aversion tend to avoid risk. Risk aversion is negatively correlated with health risks such as smoking, heavy drinking, obesity, and the non-use of seat belts [11]. In breast cancer screening, people with high-risk aversion are more likely to undergo health check-ups to avoid the health-threatening risk of breast cancer. That is, people with relatively higher adverse risk aversion are more likely to adopt healthy behaviors [12]. One way to evaluate risk aversion is by absolute risk aversion [13]. Nevertheless, the relationship between absolute risk aversion and willingness to undergo regular dental check-ups remains unknown.

We hypothesized that the health belief model and absolute risk aversion have impacts on willingness to undergo dental check-ups. The purpose of the present study was to clarify the associations between the willingness to undergo regular dental check-ups and oral health behaviors, the health belief model, and absolute risk aversion in Japanese university students.

## 2. Materials and Methods

### 2.1. Study Population

In this cross-sectional study, data were collected from individuals who had an oral examination and volunteered to participate in the study at Okayama University in April 2020. Inclusion criteria were those who agreed to participate in the study after informed consent, completed the questionnaire, and underwent the oral examination. The exclusion criteria were those who did not agree to participate in the study and provided incomplete questionnaires.

### 2.2. Ethical Procedures and Informed Consent

Informed consent was obtained verbally from all participants before the start of the study. The present study was approved by the Ethics Committee of Okayama University Graduate School of Medicine, Dentistry, and Pharmaceutical Sciences (no. 1060). This study conformed with the Strengthening the Reporting of Observational Studies in Epidemiology (STROBE) guidelines.

### 2.3. Questionnaire

The questionnaires were edited in Japanese and made by Google Forms or printed. The questionnaire included age, sex, oral health behaviors, self-rated oral health status, self-rated health status, health belief model, risk aversion, and willingness to undergo regular dental check-ups (17 items). All students completed the questionnaires at Okayama University. There was no time limit to complete the questionnaires.

#### 2.3.1. Oral Health Behaviors

Questions about oral health behaviors were as follows [14,15,16,17].

1. Are you using an interdental brush or dental floss? (Use of an interdental brush and dental floss) (Yes/No).

2. Have you been to a dental clinic for a regular dental check-up in the last year? (Dental consultation within the past year) (Yes/No).

3. Do you have a habit of eating sweet foods and beverages as a snack? (Regular snack and soft drink intake) (No/Once a day/Twice a day/Three or more times a day).

4. How many times a day do you brush your teeth?

[Tooth brushing frequency (times/day)] (Once or less/Twice/Three or more times).

#### 2.3.2. Self-Stated Oral Health Status and Self-Rated Health Status

Questions about self-rated oral health status and self-rated health status were as follows [18].

1. In general, how do you consider your oral health? (self-rated oral health status) (Very good/Good/Fair/Poor/Very poor).

2. In general, how do you consider your health? (self-rated health status) (Very good/Good/Fair/Poor/Very poor).

#### 2.3.3. Health Belief Model

Questions about the health belief model were shown using a Likert scale of 1–5 (1 = strongly disagree, 2 = disagree, 3 = neutral, 4 = agree, and 5 = strongly agree) as follows [5,19].


*Perceived susceptibility:*


I feel that I will get dental diseases within a year.


*Perceived benefits:*


1. Undergoing a dental check-up can detect dental diseases.

2. If I do not undergo a dental check-up, I am afraid that the risk of having a dental disease will remain.

3. If an oral disease is found at a dental check-up, the prognosis may be good.


*Perceived severity:*


1. If I had a dental disease, it could be detected at an early stage.

2. If I had a dental disease, it could be detected at an advanced stage.


*Perceived barrier:*


When I do not undergo a dental check-up, I do not worry as much about having an oral disease.

#### 2.3.4. Absolute Risk Aversion

The following scenario was presented, and questions were asked regarding the degree of risk aversion [20,21,22].

Please assume that you are buying a lottery ticket. There is a lottery with a 50% chance of winning 100,000 Japanese yen. What is the maximum amount of money that you would pay for this this lottery ticket?

Risk aversion was evaluated by substituting the answer amount of the question into the absolute risk aversion Formula (1).


(1)
$$\mathrm{Absolute}\mathrm{risk}\mathrm{aversion}=(\alpha Z-p)/[(1/2)({\alpha Z}^{2}-2\alpha Zp+p^{2})]$$


α = 50% chance, Z = 100,000 Japanese yen, αZ = expectation of lottery, p = maximum amount the respondent can pay to buy this lottery ticket.

If the value of absolute risk aversion is large, risk aversion is high.

#### 2.3.5. Willingness to Undergo Regular Dental Check-Ups

Questions about willingness to undergo regular dental check-ups were as follows.

We conduct dental check-ups for current students every year. Do you plan to continue undergoing dental check-ups in the future?

1. I will undergo one every year.

2. I will undergo one before graduation.

3. Not every year, but I will undergo one if I have time.

4. I will undergo one while I’m in school.

Answer 1 was classified as “willing to undergo a regular dental check-up”, and answers 2 to 4 were classified as “not willing to undergo a regular dental check-up”.

### 2.4. Oral Examination

Seven dentists (N.T., D.E., T.M, A.Y, D.F, Y.U-F, and M.N) examined oral status. The decayed, missing, and filled teeth (DMFT) score was used to evaluate dental caries status based on the World Health Organization caries diagnostic criteria [23]. The percentage of bleeding on probing in ten teeth was assessed as an indicator of inflammation [2]. The areas examined were two molars in each posterior sextant and the upper right and lower left central incisors. The level of dental plaque and calculus was assessed using the Oral Hygiene Index-Simplified (OHI-S). The areas examined were the buccal aspect of the upper first molars, the upper right central incisor, the lower left central incisor, and the lingual aspect of the lower first molars [24]. After training the examiners, they checked the DMFT and OHI-S scores in two volunteers for two weeks. For the oral examination, intra- and inter-agreements were good (Kappa statistic > 0.8).

### 2.5. Sample Size

Since there were no previous reports that investigated the oral health behaviors, the health belief model, and absolute risk aversion on the willingness of Japanese university students to undergo regular dental check-ups, sample size estimation was not performed. Thus, all data were included.

### 2.6. Bias

All participants who matched the inclusion criteria were included to minimize selection bias.

### 2.7. Statistical Analysis

Statistical analyses were conducted using SPSS version 22 (IBM, Tokyo, Japan). Values of *p* < 0.05 were considered to indicate significant associations.

The Mann–Whitney *U* or chi-squared test was used to compare students who were and were not willing to undergo regular dental check-ups. Logistic regression analysis was conducted to investigate the associations between willingness for regular dental check-ups and independent variables. Independent variables were selected when their *p* values were <0.20 on the chi-squared or Mann–Whitney *U* test, because, based on previous studies, it has been suggested that potential confounders should be eliminated only if *p* > 0.20 to prevent residual confounding [25]. 

## 3. Results

Figure 1 shows the flowchart of this study. The response rate was 81.0%.

Table 1 shows the associations between willingness for regular dental check-ups and the factors examined. A total of 285 students were willing to undergo regular dental check-ups. The range of age was 18 to 35 years old. Willingness was significantly associated with 12 factors (*p* < 0.05), but not with age, absolute risk aversion, DMFT, BOP, and OHI-S.

Table 2 shows the odds ratios of factors associated with willingness to undergo regular dental check-ups. Willingness to undergo regular dental check-ups was associated with the use of interdental brushes and dental floss (odds ratio, 1.616; 95% CI, 1.104–2.365; *p* < 0.001) and perceived benefits—2 (If I do not undergo a dental check-up, I am afraid that the risk of having a dental disease will remain. (odds ratio, 1.683; 95% CI, 1.254–2.260; *p* < 0.001). Those results showed that subjects who were willing to undergo regular dental check-ups used interdental brushes and dental floss more and recognized the risk factors of dental disease.

## 4. Discussion

To the best of our knowledge, the present study is the first to investigate the associations between willingness to undergo regular dental check-ups and oral health behaviors, the health belief model, and risk aversion. Since dental check-ups play an important role in maintaining good oral health [26], investigating factors associated with willingness to undergo regular dental check-ups may contribute to increasing regular dental check-ups and maintaining good oral health.

Willingness to undergo regular dental check-ups was significantly associated with one of the perceived benefits categories in the health belief model, i.e., “If I do not undergo a dental check-up, I am afraid that the risk of having a dental disease will remain.” (Table 2). Perceived benefits construct parts of the health belief model that refer to one’s beliefs about the benefit of recommended behaviors in reducing the risk of a disease or its consequences [27]. In past studies, perceived benefits have been shown to be directly associated with colorectal cancer screening intention or behavior [28,29,30]. This suggested that individuals are more likely to perform a preventive health behavior when they perceive themselves to be at risk of a negative health outcome, and they can see a benefit to performing the recommended health behavior [31,32]. The present study also suggests that the preventive health behavior of a dental check-up may have been taken to prevent the negative outcome of the risk of dental disease due to skipping a dental check-up.

Willingness to undergo regular dental check-ups was not significantly associated with the other categories in the health belief model (Table 2). Past studies showed that fears of needles or dental injections are potential barriers to one of the health belief models, leading to poor oral health and utilization of dental care [8]. Another study showed that perceived susceptibility and seriousness were not significantly associated with the performance of breast self-examination (BSE) [33]. Younger individuals did not think that they were at risk of breast cancer and thus did not independently pursue information about BSE [33,34]. Another explanation might be the lack of sufficient knowledge about breast cancer in this young age group [33] Thus, susceptibility and seriousness might be low for breast cancer. Similarly, the participants in the present study were young. Therefore, they may lack knowledge about dental disease and may not think of the risks.

Willingness to undergo regular dental check-ups was significantly associated with the use of interdental brushes or dental floss (Table 2). The actual reason for the association is unclear. However, self-efficacy might affect the association. Self-efficacy is advocated within the framework of social learning theory and is defined as an individual’s confidence in determining “how well he or she can take the actions necessary for producing certain results” [35]. Self-efficacy was associated with the frequency of flossing [36]. Another study showed that people using interdental brushes had high self-efficacy scores [37]. Higher self-efficacy scores might motivate students to progress through changes in oral health behaviors [37]. In this study, participants who used dental floss or interdental brushes might have had higher self-efficacy and been willing to undergo dental check-ups.

There was no significant association between willingness to undergo regular dental check-ups and absolute risk aversion (Table 2). Past studies suggested that young people had lower risk aversion, and they developed higher risk aversion as they grew older [38,39]. Since the participants in the present study were limited to young people (mean age 18.5 years), their absolute risk aversion may have been lower. The absolute risk aversion in this study was 2.0 × 10^−5^ and was lower than that (1.8 × 10^−3^) of the past study, where the group had a mean age of 52.1 years [22]. The discrepancies between the results of the present study and those of past studies may be explained by differences in the targeted age groups. It is also known that there are differences in risk aversion depending on race. However, the present study was limited to the Japanese population and did not need to consider the differences in risk aversion by race [40]. Since there are few past studies investigating the relationship between absolute risk aversion and dentistry, it is difficult to compare with past studies. Further research is needed to investigate the relationship between these factors.

Oral health status (DMFT, %BOP, and OHI-S) was not associated with willingness to undergo regular dental check-ups. Oral health status in the present study was better overall than in recent studies of university students (DMF 0.02 vs. 1.6, %BOP 16.7% vs. 35.5%, and OHI-S 0.2 vs. 0.5) [3,41]. The participants might have been more interested in oral health and had good oral health behaviors. If students have poor oral health status, it can easily induce oral symptoms and motivate students to go to dental clinics. The present study may not have found an association because the students did not have poor oral health status or there may have been floor effects. 

Several limitations of the present study must be considered when interpreting the results. First, the findings may not be generalizable to other young people. The participants were from Okayama University, which may limit the generalizability of the findings. Second, this investigation was a cross-sectional study that can only determine associations between variables; it is not capable of examining cause-and-effect relationships between the variables. Third, only 10 teeth were examined in the BOP and 6 teeth were examined in oral hygiene status, which might have led to under- or overestimation.

## 5. Conclusions

The willingness of Japanese university students to undergo regular dental check-ups was associated with oral health behaviors and the health belief model, but not absolute risk aversion.

## Figures and Tables

**Figure 1 ijerph-19-13920-f001:**
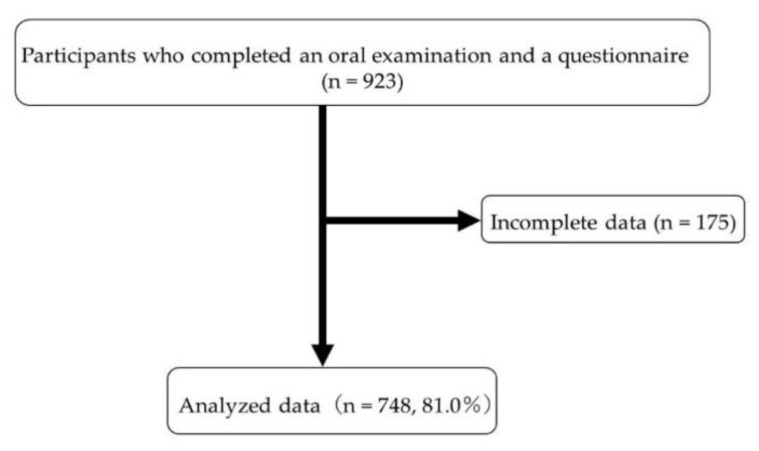
Flowchart of the complete data, excluding incomplete data, of those who completed the oral examination questionnaire.

**Table 1 ijerph-19-13920-t001:** Associations between willingness for regular dental check-ups and related factors.

Factors		Willingness		*p* ^‡^
		(+) (*n* = 285)	(−) (*n* = 463)	
Age		18.0 (18.0, 19.0) *	18.0 (18.0, 18.0)	0.489
Male		123 (33.3) ^†^	246 (66.7)	0.008
Oral health behaviors				
Use of the interdental brush or the dental floss	No	198 (34.3)	380 (65.7)	<0.001
	Yes	87 (51.2)	83 (48.8)	
Dental consultation within the past year	No	111 (31.3)	244 (68.7)	<0.001
	Yes	174 (44.3)	219 (55.7)	
Snack and soft drink intake regularly	No	79 (42.5)	107 (57.5)	0.091
	Once a day	164 (37.8)	270 (62.2)	
	Twice a day	31 (31.0)	69 (69.0)	
	Three times or more a day	11 (39.3)	17 (60.7)	
Tooth brushing frequency (times/day)	One time or less	14 (26.9)	38 (73.1)	0.015
	Twice	186 (36.9)	318 (63.1)	
	Three times or more	85 (44.3)	107 (55.7)	
Self-rated oral health status	Very poor	7 (38.9)	11 (61.1)	0.013
	Poor	37 (34.6)	70 (65.4)	
	Fair	114 (33.6)	225 (66.4)	
	Good	95 (44.6)	118 (55.4)	
	Very good	32 (45.1)	39 (54.9)	
Self-rated health status	Very poor	4 (57.1)	3 (42.9)	0.020
	Poor	7 (24.1)	22 (75.9)	
	Fair	47 (32.2)	99 (67.8)	
	Good	133 (38.1)	216 (61.9)	
	Very good	94 (43.3)	123 (56.7)	
Health belief model				
Perceived susceptibility		3.0 (2.0, 3.0)	3.0 (2.0, 3.0)	0.084
Perceived benefits—1		5.0 (4.0, 5.0)	5.0 (4.0, 5.0)	0.001
Perceived benefits—2		5.0 (4.0, 5.0)	4.0 (4.0, 5.0)	<0.001
Perceived benefits—3		5.0 (4.0, 5.0)	5.0 (4.0, 5.0)	<0.001
Perceived severity—1		4.0 (3.0, 5.0)	3.0 (3.0, 4.0)	0.001
Perceived severity—2		3.0 (2.0, 4.0)	3.0 (3.0, 4.0)	0.005
Perceived barrier		2.0 (1.0, 3.0)	2.0 (1.0, 3.0)	<0.001
Absolute risk aversion (×10^−5^)		2.0 (1.8, 2.0)	2.0 (1.8, 2.0)	0.253
Oral health status				
Decayed, missing, and filled teeth		0.000 (0.000, 0.003)	0.000 (0.000, 0.003)	0.698
Percentage of bleeding on probing		10.0 (0.0, 30.0)	10.0 (0.0, 30.0)	0.135
Oral Hygiene Index-Simplified		0.2 (0.0, 0.7)	0.3 (0.0, 0.7)	0.093

* Median (25%tile, 75%tile); ^†^ Number of people (%); ^‡^ Chi-square test or Mann–Whitney *U* test.

**Table 2 ijerph-19-13920-t002:** Factors associated with willingness to undergo regular dental check-ups (logistic regression analysis).

		OR *	95% CI	*p*
Gender	Male	1	0.865–1.655	0.279
	Female	1.196		
Oral health behaviors				
Use of the interdental brush or the dental floss	No	1		
	Yes	1.616	1.104–2.365	0.014
Dental consultation within the past year	No	1		
	Yes	1.371	0.985–1.909	0.061
Tooth brushing (times/day)	One time or less	1		
	Twice	0.923	0.639–1.333	0.669
	Three times or more	0.663	0.319–1.376	0.270
Snack and soft drink intake regularly	No	1		
	Once a day	0.834	0.332–2.099	0.700
	Twice a day	0.978	0.425–2.248	0.958
	Three times or more a day	1.225	0.515–2.915	0.646
Self-rated oral health status		1.007	0.805–1.260	0.948
Self-rated health status		1.083	0.885–1.341	0.417
Health belief model				
Perceived susceptibility		0.942	0.784–1.131	0.519
Perceived benefits—1		1.175	0.906–1.522	0.224
Perceived benefits—2		1.683	1.254–2.260	<0.001
Perceived benefits—3		1.044	0.740–1.474	0.806
Perceived severity—1		1.167	0.975–1.397	0.092
Perceived severity—2		0.917	0.774–1.068	0.316
Perceived barrier		0.922	0.796–1.068	0.279
Oral Hygiene Index-Simplified		0.817	0.562–1.187	0.288

* Adjusted odds ratio; OR, odds ratio; CI, confidence interval.

## Data Availability

All the relevant data are included in the manuscript.

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
