# Peer review of "The Impact of Oral Health Behaviors, Health Belief Model, and Absolute Risk Aversion on the Willingness of Japanese University Students to Undergo Regular Dental Check-Ups: A Cross-Sectional Study"

_ijerph, 2022, doi:10.3390/ijerph192113920_

Round 1
Reviewer 1 Report
This is a well done study and clarify the associations between the willingness to undergo regular dental check-ups and oral health behaviors, health belief model, and absolute risk aversion in Japanese university students. In spite of the good work done, I suggest the following changes in order to improve this paper further:
Materials and Methods
1. The inclusion criteria and exclusion criterion are too simple. Please explain more thoroughly.
2. In the section of oral examination, did you have the results of the inter-examiner reliability?
3. How was your sample size calculated? The reason you mentioned seems unreasonable.
4. How did you choose the confounders? Please give details on your choice.
Author Response
Reviewer 1
This is a well done study and clarify the associations between the willingness to undergo regular dental check-ups and oral health behaviors, health belief model, and absolute risk aversion in Japanese university students. In spite of the good work done, I suggest the following changes in order to improve this paper further:
Our comments: Thank you for your comments. We have revised our manuscripts based on your comments.
Materials and Methods
- The inclusion criteria and exclusion criterion are too simple. Please explain more thoroughly.
Our comments: Thank you for your comment. We have added explanation following the reviewer’s suggestion. (lines 72-75).
- In the section of oral examination, did you have the results of the inter-examiner reliability?
Our comments: Thank you for your comment. After training the examiners, they checked the DMFT and OHI-S score in two volunteers for two weeks. For the oral examination, intra- and inter-agreements were good (Kappa statistic > 0.8) We have added the sentences following the reviewer’s suggestion (lines 152-154).
- How was your sample size calculated? The reason you mentioned seems unreasonable.
Our comments: Thank you for your comment. Since there were no previous reports that investigated the oral health behaviors, health belief model, and absolute risk aversion on the willingness to undergo regular dental check-ups, sample size estimation was not per-formed. Thus, all data were included. We have revised the manuscript following the reviewer’s suggestion (lines 156-159).

- How did you choose the confounders? Please give details on your choice.
Our comments: Thank you for your comment. We selected confounders which were the value of P <0.20 in Mann-Whitney U or chi-squared test. (lines 166-172)
Reviewer 2 Report
Congratulations to the authors for conducting the research described in the article “The impact of oral health behaviors, health belief model, and absolute risk aversion on the willingness of Japanese university students to undergo regular dental check-ups: a cross-sectional study”.
The manuscript is well-written and only requires minor changes.
Please see my comments below.
Introduction
· The Introduction does not require any additional changes. It offers enough information regarding the topic of this paper, and the aim is clearly described at the end of the section.
Materials and methods
· Lines 81-83 contain some general information regarding the questionnaire. “The questionnaire included age, sex, oral health behaviors, self-rated oral health status, self-rated health status, health belief model, risk aversion, and willingness to undergo regular dental check-ups.” I recommend adding at the end of this phrase the total number of items used in the questionnaire.
· Please improve this section by adding the answers to the following questions:
a. What was the language used in the questionnaires?
b. Where were the questionnaires completed?
c. Were the questionnaires printed?
d. Was there a time limit imposed for completing the questionnaires?
e. What was the age range of the participants?
Results
· The information contained in the Results section, both in the text and the two tables, is enough, and well described.
Discussion
· The section generously presents the topic and compares it with other similar research. The limitations of the research are also well described.
Conclusions
· The section is concise, and I recommend not making any additional changes.
Best regards!
Author Response
Reviewer 2
Congratulations to the authors for conducting the research described in the article “The impact of oral health behaviors, health belief model, and absolute risk aversion on the willingness of Japanese university students to undergo regular dental check-ups: a cross-sectional study”.
The manuscript is well-written and only requires minor changes.
Please see my comments below.
Our comments: Thank you for your comments. We have revised our manuscripts based on your comments.
Introduction
- The Introduction does not require any additional changes. It offers enough information regarding the topic of this paper, and the aim is clearly described at the end of the section.
Our comments: Thank you for your comment.
Materials and methods
- Lines 81-83 contain some general information regarding the questionnaire. “The questionnaire included age, sex, oral health behaviors, self-rated oral health status, self-rated health status, health belief model, risk aversion, and willingness to undergo regular dental check-ups.” I recommend adding at the end of this phrase the total number of items used in the questionnaire.
Our comments: Thank you for your comment. We have added the number of question items (17 items) following the reviewer’s suggestion (line 86).
- Please improve this section by adding the answers to the following questions:
a. What was the language used in the questionnaires?
Our comments: Thank you for your comment. The questionnaires were edited in Japanese. We have added the comment following the reviewer’s suggestion (line 83).
b. Where were the questionnaires completed?
Our comments: Thank you for your comment. The questionnaires were completed at examination room of Okayama University. We have added the place following the reviewer’s suggestion (lines 86-87).
c. Were the questionnaires printed?
Our comments: Thank you for your comment. The questionnaires were disseminated on Google Forms or prints. We have added the comment following the reviewer’s suggestion (lines 83-84).
d. Was there a time limit imposed for completing the questionnaires?
Our comments: Thank you for your comment. There was no time limit to complete the questionnaires. We have added the sentence following the reviewer’s suggestion (line 87).
e. What was the age range of the participants?
Our comments: Thank you for your comment. The range of age was 18 to 35 years old. We have added the comment following the reviewer’s suggestion (line 180).
Results
- The information contained in the Results section, both in the text and the two tables, is enough, and well described.
Discussion
- The section generously presents the topic and compares it with other similar research. The limitations of the research are also well described.
Conclusions
- The section is concise, and I recommend not making any additional changes.
Best regards!
Our comments: Thank you for your comments.